# A Method of Personalized Driving Decision for Smart Car Based on Deep Reinforcement Learning

**Xinpeng Wang [1]**, **Chaozhong Wu [1]**, **Jie Xue [1,2]** and **Zhijun Chen [1,*]**

[1]  Intelligent Transportation Systems Center (ITSC), Wuhan University of Technology, Wuhan 430000, China; xp_wang@whut.edu.cn (X.W.); wucz@whut.edu.cn (C.W.)

[2]  Faculty of Technology, Policy and Management, Safety and Security Science Group (S3G), Delft University of Technology, 2628BX Delft, The Netherlands; J.Xue@tudelft.nl

[*]  Correspondence: chenzj556@whut.edu.cn; Tel.: +86-158-2712-7655

**Abstract:** To date, automatic driving technology has become a hotspot in academia. It is necessary to provide a personalization of automatic driving decision for each passenger. The purpose of this paper is to propose a self-learning method for personalized driving decisions. First, collect and analyze driving data from different drivers to set learning goals. Then, Deep Deterministic Policy Gradient algorithm is utilized to design a driving decision system. Furthermore, personalized factors are introduced for some observed parameters to build a personalized driving decision model. Finally, compare the proposed method with classic Deep Reinforcement Learning algorithms. The results show that the performance of the personalized driving decision model is better than the classic algorithms, and it is similar to the manual driving situation. Therefore, the proposed model can effectively learn the human-like personalized driving decisions of different drivers for structured road. Based on this model, the smart car can accomplish personalized driving.

**Keywords:** smart car; personalization; driving decision; human-like; deep reinforcement learning; data visualization

## 1. Introduction

With the advancement of science and technology, especially the rapid development of Internet technology, communication technology and artificial intelligence technology in recent years [1], the era of smart car technology has come. However, the practical use of smart cars requires a long transition period. Despite the tremendous development of smart car technology, there are still some issues that need to be addressed. The driving environment has complexity and variability, which brings great challenges to driving decisions. In addition, the smart car automatic driving system is generally designed based on fixed rules that do not consider the driver's personalization of driving. This seriously hinders the development and practicality of smart car driving technology. Therefore, conducting human driving research can effectively solve the driving decision problem of smart cars.

Usually, automatic driving systems involve three steps: perception, decision, and control [2]. The setting of this system architecture is consistent with human driving behavior. The smart car first obtains sensor information through various sensors, fuses different sensor information, forms multienvironment information, and transmits it to the planning decision-making layer [3]. Then, according to the constraints and rules set by the system, the decision is made to generate corresponding driving operation information. Finally, the control module executes commands to achieve automatic driving. Among them, the decision-making link enables the smart car to map the environmental perception of the driving action, which is a key aspect of automatic driving technology.

There are generally two methods for implementing driving decisions. One is based on rules, and the other is based on algorithms. Rule-based decision making first builds a library of driving behavior rules based on traffic regulations, driving rules, and driving experience. According to the rule base, the driving behavior of the smart car is divided, and then the state of the smart car is divided according to the driving environment. Then, the smart car driving behavior is generated according to the rule logic. In other words, some colleges and universities cooperate with enterprises to develop smart car decision-making systems using classic methods and achieve certain results [4–6]. Okumura et al. proposed a driving decision method that can be incorporated into a roundabout [7]. They used learning from demonstration to construct a classifier for high-level decision making and developed a novel set of formulations suited to this challenging situation: multiple agents in a highly dynamic environment with interdependencies between agents, partial observability, and a limited quantity of training data.

With the advancement of machine learning technology, some researchers have applied relevant technologies in driving decision research. NVIDIA released a driving decision algorithm that trains a Convolutional Neural Network (CNN) to build the relationship between environmental information and vehicle control, enabling "end-to-end" control [8]. The network input is the information of the car and the environmental information captured by the camera. The output is the direct control of the vehicle, that is, the brake, the throttle, and the steering. This method omits the middle layer and directly establishes state-operation mapping.

Chen et al. proposed a learning method for intelligent car driving decision-making based on CNN for the highly structured driving environment of highways [9]. Unlike NVIDIA, the input image is not directly mapped to the execution action of the control vehicle. The relationship between the input image and a series of key perceptual indicators (such as vehicle position and attitude, current road and traffic status) is intermittently established and the execution action is determined based on the perceptual indicators. The author maps the original input image to the driving environment state characterizer. The 13 parameters distinguish between the two conditions of lane keeping and driving state change. According to different characterization combinations, driving decisions of steering and acceleration and deceleration are generated. The results of testing using the KITTI (Karlsruhe Institute of Technology and Toyota Technological Institute) data set show that the model performs better than the previously proposed method in both virtual and real structured roads.

German BMW and Munich University of Technology proposed a decision model based on a partially observable Markov decision process, which mainly solves the decision problem in dynamic and uncertain driving environments [10]. The uncertainty is mainly due to the uncertainty of sensor noise and the driving intention of traffic participants. POMDP uses the driving intentions of other vehicles as hidden variables to establish a Bayesian probability model to solve the optimal acceleration of the vehicle on the planned path. Tan proposed an automatic driving system based on deep reinforcement learning [11]. The system uses four consecutive frames of semantic segmentation images as input parameters. The gap between the simulation platform and the actual road conditions decreased and it achieved good decision-making results.

Loiacono et al. designed a driving decision model with Q-learning algorithm as the core for overtaking scenarios on structured roads [12]. When overtaking, fully consider the obstacle avoidance requirements and lane line position, generate acceleration and deceleration decisions and overtaking trajectory, and achieve the purpose of overtaking by controlling the actuator of the virtual vehicle.

Traditional automatic driving technology research usually acquires sensor signals, translates them into a human-understandable driving environment, and constructs an expert rule system based on human manual driving rules, which corresponds to the vehicle control action end-to-end. However, the reality of the driving environment is complex and variable. It is not easy to abstract driving rules into simple logic or mathematical formulas. Therefore, many classic research methods cannot comprehensively and effectively address complex traffic environments. Currently, automatic driving techniques are generally designed based on collision avoidance rules, traffic rules, and driving decisions of drivers with fixed benchmarks. The differentiation of driving habits is not considered. Driving

with different driving habits has a greater impact on passenger comfort. The same driving decision cannot meet the personalized needs of different passengers. This seriously hinders the development and practical use of automatic driving technology.

Usually, the formation of driving habits is influenced by many factors, and driving habits also contain features of multiple dimensions [13]. Traditional methods cannot effectively extract the driver's complete driving habits. Driving habits are reflected in the details of all driving operations during daily driving. The driving strategy can be obtained using relevant parameters. However, in the daily driving process, a large number of structured datasets are generated. There are complex correlations between different data, and methods of manual extraction and analysis cannot be effectively dealt with. Reinforcement learning has shown good learning performance in strategy learning. At present, deep reinforcement learning is widely used in the simulation, industrial control and gaming [14]. Among them, the Deep Deterministic Policy Gradient (DDPG) algorithm has good convergence and stability [15,16]. The algorithm offers possibilities for automatic driving research.

This paper proposes a self-learning method for personalized driving decisions based on deep reinforcement learning. First, different drivers are invited to conduct driving experiments. The driving data are collected for comparative analysis, the characteristics of personalized driving decisions are mined, and the selection method of personalized driving indicators is determined. Then, the driving decision learning model is designed based on the simulation platform, the input and output are determined, and a reasonable reward function is set. The online interactive training model is adopted in the simulation platform. After the model converges, the introduction of personalization factors is associated with key indicators. Factor values are adjusted, and personalized driving strategies are learned, which are tested on different maps. Finally, the personalized driving data are compared and analyzed, and the effect of personalized decision learning is discussed.

## 2. Materials and Methods

To realize the self-learning of personalized driving decisions, a personalized driving decision model needs to be built for smart cars. First, how to embody driving personalization needs to be determined. A driving experiment is used to collect driving data to determine the driving decision rules of different drivers and to develop a personalized expression. Then, a decision-making system that can autonomously learn a safe driving strategy is designed. Finally, the personalized factors are integrated into the decision-making system so that it can use the personalized differences in the automatic driving phase.

### 2.1. Personalized Driving Indicator Selection

Three types of drivers are invited to participate in the manual driving experiments [17]. The driving data is collected and their differences during driving are analyzed. Some indicators are identified to reflect driving personalization. Manual driving data are the target and evaluation criteria for learning the personalized driving decision model. Our ultimate goal is that the model's driving data are similar to different styles of driver data.

Simulation software is used to simulate driving scenarios. To verify the universality of the proposed method, we choose the UC-win/ROAD platform for manual driving experiments [18]. A total of 11 parameters are output (Table 1), including 8 vehicle motion parameters and 3 action parameters.

**Table 1.** Manual driving acquisition parameters.

| Parameter Name | Interpretation |
| --- | --- |
| **Vehicle Motion Parameters** | |
| angle | Angle between the vehicle direction and the direction of the track |
| damage | Current damage of the vehicle |
| rpm | Number of rotation per minute of the car engine |
| speed X | Speed of the car along the longitudinal axis of the vehicle |
| speed Y | Speed of the car along the transverse axis of the vehicle |
| track Dis | Vehicle distance from track edge |
| track Pos | Distance between the car and the track axis |
| wheel Spin Vel | Vector of 4 sensors representing the rotation speed of the wheels |
| **Driving ActionParameters** | |
| accel | The throttle opening (0 means no gas, 100% full gas) |
| steering | Steering value: −1 and +1 means respectively full left and right |
| brake | Brake level (0 means no brake, 100% full brake) |

## 2.1.1. Manual Driving Scene Setting

This research is oriented toward structured roads. UC-win/ROAD simulates an urban driving environment. The platform scene setting is more realistic, which gives the driver a greater sense of driving immersion. The driving environment is set to six two-way lanes. A total of 20 other moving vehicles, which controlled by the platform, are added to the road. This allows the driver to perform driving actions such as acceleration, deceleration, and steering. The speed of the other vehicles fluctuates within 10%. To increase the difficulty of driving and to ensure that the driving scene is not too complicated, the lane change probability of other vehicles is set to 5%. The initial speed of the vehicle is set to 15 m/s to avoid being rear-ended by other vehicles in the initial stage. The length of a single manual driving section is limited to 4 km. There is a prompt every time the drive reaches 4 km, and the platform no longer collects data. When the vehicle collides or brakes hard, the driving session is terminated.

A total of 30 drivers were invited to participate in the experiment. They all had proficient driving skills. The drivers were required to drive the vehicle in the center of the lane and avoid collisions with other vehicles. Each driver drove for 4 rounds. Excluding collisions and undesired processes, a total of 25 drivers' data were retained.

## 2.1.2. Manual Driving Analysis

Previous research has not formed a unified indicator on the individualized expression of driving decisions [19,20]. Various parameters of different driving style data have been analyzed, and there are large differences in speed, throttle opening, and steering. Therefore, this paper chooses these three parameters as the expression of personalized driving decisions. The two speed components are combined into a vector. The throttle opening represents an action to increase the vehicle speed, which is a parameter of the degree of depression of the accelerator pedal. Although the speed parameter is related to the throttle parameter, it is not a simple linear relationship, and the contents of the responses are different [21].

Analysis of the same driving style data shows that drivers perform similarly on the three parameters [22]. Therefore, one driver is selected as an example for each style to analyze the differences in different driving styles (Figure 1). Complete data statistics are in the experimental results section. We divide the data into spans of 150 s, during which the drivers complete 2 complete acceleration, deceleration and steering processes.

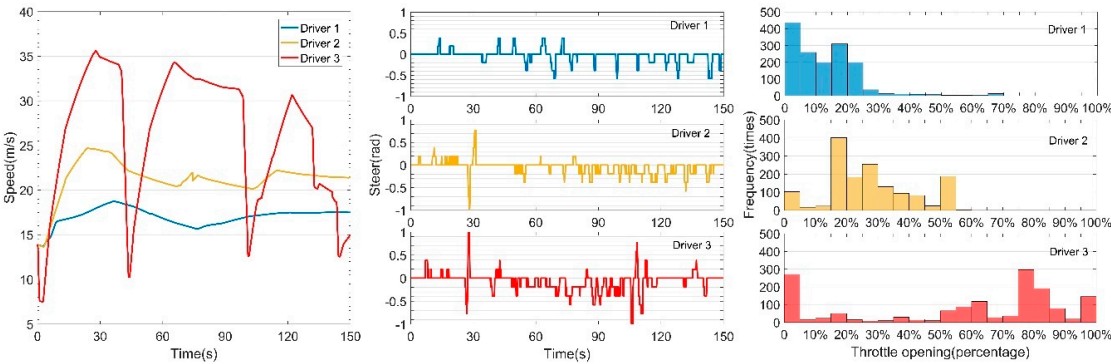

**Figure 1.** Examples of driving data in different styles.

In terms of driving speed, Driver 1 had a relatively low speed during driving, and the speed was very stable. The maximum speed of Driver 3 was significantly higher than that of other drivers, and the fluctuation was extremely high. In terms of steering, the data distribution of the three drivers was not much different. In terms of throttle opening, the data of Driver 1 were relatively flat. The throttle opening of Driver 3 was relatively high, and the throttle may have been fully opened.

In similar driving situations, the throttle opening and steering operation frequency of Driver 1 were lower and to a lesser extent. Driver 1 represents a conservative driving style. The throttle opening and steering operation frequency of Driver 3 were relatively high, and the degree was higher. The maximum speed was exhibited, and the fluctuation was high. Driver 3 represents an aggressive driving style. The data of driver 2 showed an intermediate driving style. This is consistent with previous research and our analysis. The design and evaluation criteria for personalized driving decisions considered the differences between the three driving styles in terms of speed, steering and throttle opening.

### 2.2. Driving Decision System Based on Deep Reinforcement Learning

### 2.2.1. Reinforcement Learning

Reinforcement learning (RL) evolved from theories of animal learning and adaptive control of parameter perturbations. Emphasis is a self-learning process from environmental state to action mapping. The agent interacts with the environment at each moment and selects actions. The environment reacts to this action and reaches a new state, then evaluates the behavior through a value function, and finally determines the optimal strategy to reach the target state [23]. In general, reinforcement learning places the agent in the environment to perform actions and learns strategies by rewarding. This type of training requires the network to work simultaneously with the environment.

Q-learning is a classic model-free reinforcement learning algorithm. A two-dimensional table is constructed to represent its policy hypothesis $\hat{Q}$. Each state-action pair has an entry $\hat{Q}(s, a)$. The agent observes the current state $s$, selects an action $a$ and executes it, and then observes the return value $r$ and the next state $s'$, where $\gamma$ is the discount factor. The agent repeats this process and updates $\hat{Q}(s, a)$ with Equation (1) [24].

$$\hat{Q}(s, a) \leftarrow r + \gamma \max_{a'}\hat{Q}(s', a') \; 0 \leq \gamma \leq 1 \tag{1}$$

With the development of machine learning, reinforcement learning and deep learning are combined to form new methods that combine their respective advantages [25]. This creates a new popular field of artificial intelligence. Deep reinforcement learning combines the perceptual power of deep learning with the decision-making ability of reinforcement learning in a common form. Direct control from raw input to output can be achieved through end-to-end learning. Traditional reinforcement learning algorithms cannot work in a continuous action domain and can only discretize output actions. However, discretizing the driving action has lost its application significance and cannot provide a

comfortable environment for passengers. Then, the Deep Deterministic Policy Gradient (DDPG) algorithm [26] was proposed by DeepMind. The algorithm uses a convolutional neural network as a simulation of the strategy function and the action-value function. The algorithm works in a continuous action domain and does not require a model. The algorithm is improved by the Deterministic Policy Gradient (DPG) algorithm based on the actor-critic framework.

### 2.2.2. Driving Decision System Design

The driving decision system (Figure 2) is designed based on the DDPG algorithm. The deep learning part follows the network structure of the algorithm. The number of neurons in the input and output layers changes according to the system design. The algorithm performs fitting learning on two functions at the same time. The critic network approximates the value function $Q : S \times A \rightarrow R$. It estimates the value $q$ of the expected discount bonus when using action $a$ in state $s$. The actor network approximates the strategy function $\pi : S \rightarrow A$. With the input determined, the best strategy (Equation (2)) is estimated and the best action is selected.

$$\pi(s) = arg \max_a q \tag{2}$$

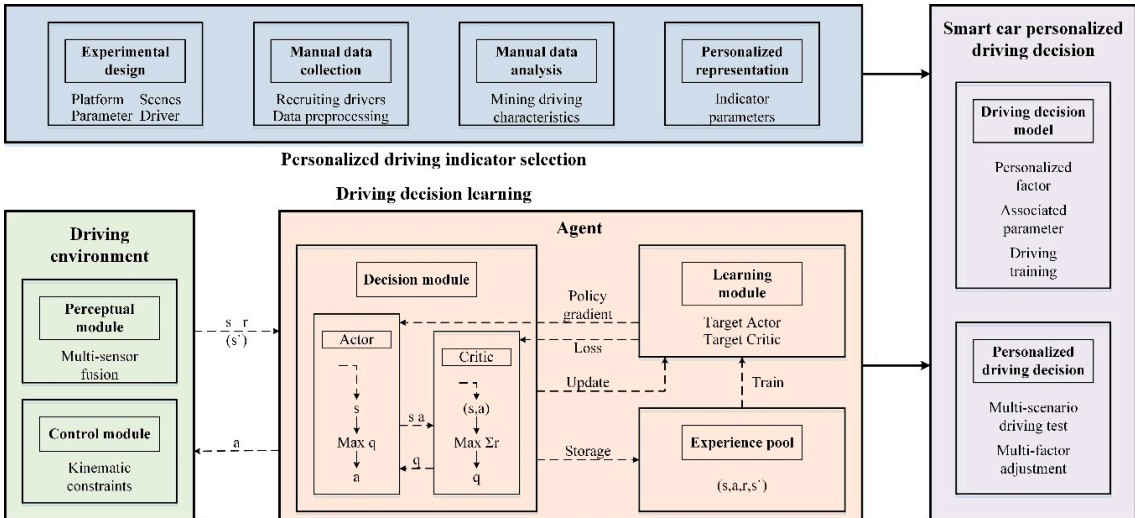

**Figure 2.** Structure of driving decision system.

After the agent interacts with the environment, the state, action, reward value, and next state are stored in the experience pool in the form of $(s_t, a_t, r_t, s_{t+1})$. When the experience reaches a certain number, random samples are generated to be trained by the algorithm.

In reinforcement learning, the reward function (Equation (3)) has a greater impact on learning outcomes. The model obtains the sensor parameters, determines the behavior that should be taken, reaches a new state and obtains the corresponding reward. To ensure that the driving operation of the model is relatively stable, the vehicle speed and steering wheel speed are set to the upper limit, ensuring that the algorithm does not take unreasonable driving action.

$$\begin{cases} R_t = V\alpha \cos(\theta) - V\beta \sin(\theta) - V\delta|d|, & no\ damage \\ R_t = V\alpha \cos(\theta) - V\beta \sin(\theta) - V\delta|d| - 100, & damage \end{cases} \tag{3}$$

The parameter $V$ is the speed of the vehicle in the forward direction. The parameter $\theta$ is the angle at which the vehicle points to the tangential direction of the lane centerline. The parameter $d$ is the offset of the vehicle center point from the lane centerline. The parameters $\alpha, \beta$ and $\delta$ are the longitudinal coefficient, the lateral coefficient, and the deviation coefficient, respectively. The model needs to reach

a certain longitudinal speed and minimize the lateral offset speed. In addition, the model needs to stay as close to the center of the lane as possible. This is highly consistent with the purpose of everyday driving. If the vehicle runs off the road, it receives additional penalties (Figure 3).

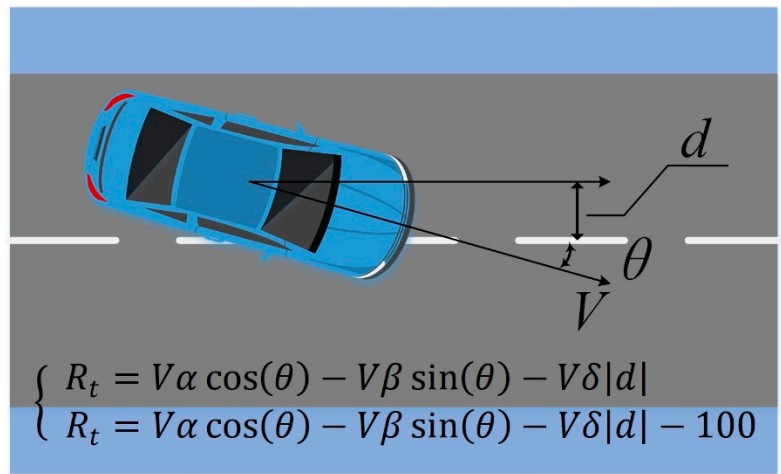

**Figure 3.** Reward function setting diagram.

In terms of constructing the state space and action space, the TORCS platform supports a variety of sensors and controller calls [27]. A detailed description of the parameters can be found in the original paper. In the driving decision system, we choose the following sensors and controllers as inputs and outputs (Table 2). We need to build a system that can safely perform driving decision tasks and prepare for a personalized driving decision model.

**Table 2.** Driving decision system input and output parameters.

| Parameter Name | Interpretation |
|---|---|
| **Input Parameters (State Space)** | |
| angle | Angle between the vehicle direction and the direction of the track |
| damage | Current damage of the vehicle |
| rpm | Number of rotation per minute of the vehicle engine |
| speed X | Speed of the car along the longitudinal axis of the vehicle |
| speed Y | Speed of the car along the transverse axis of the vehicle |
| track | Vector of 19 range finder sensors: indicates the distance of the vehicle from the edge of the track |
| track Pos | Distance between the vehicle and the track axis |
| wheel Spin Vel | Vector of 4 sensors representing the rotation speed of the wheels |
| **Output Parameters (Action Space)** | |
| accel | The throttle opening (0 means no gas, 100% full gas) |
| steering | Steering value: −1 and +1 means respectively full left and right |
| brake | Brake level (0 means no brake, 100% full brake) |

## 2.3. Personalized Driving Decision Model

In the real world, the personalization of driving decisions is diverse. Faced with the current driving environment, drivers make driving decisions based on their judgment of the environment, consideration of traffic rules, and preferences. Then, the vehicle is controlled to complete the interaction with the driving environment. The final driving behavior is related to the driver's perception of the driving environment.

Inspired by this phenomenon, coefficients are added to the sensing parameters of the decision system. Artificially disturbing the environmental perception results forms a personalized driving

decision model, the personalized driving factor—deep deterministic policy gradient (PF-DDPG). The personalization factor K is introduced into the driving decision system to build a personalized driving decision model (Equation (4)).

$$obs = damage + wheel\ Spin\ Vel + \frac{angle}{K} + \frac{rpm}{K} + \frac{speed}{K} + \frac{track}{K} + \frac{track\ Pos}{K} \tag{4}$$

In the driving decision system, the weights of the 8 sensing parameters are the same. After pre-experiments, we identified 6 sensing parameters with greater influence. Damage parameters and wheel speed parameters, which play a positive role in driving decision-making tasks, were retained. The parameter "obs" represents the state of the input. The parameter "speed" is synthesized from the speed components in Table 2. The driving decisions of the 3 driving styles were learned by changing the personalization factor. The factor values of learning conservative style, intermediate style and aggressive style in the proposed method were set to 0.47, 1.13 and 1.91, respectively.

## 3. Results

*Training and Test Results*

It is dangerous to use real vehicles on real roads to train personalized driving decision models. Damage to vehicles and roads during the initial exploration phase is incalculable. Therefore, the TORCS driving simulation platform was used for training and testing [28]. The platform was also used to verify the applicability of the proposed method to structured roads such as closed tracks. TORCS considers various factors, such as vehicle dynamics. We were able to focus on algorithmic debugging and parameter adjustment for deep reinforcement learning. The TORCS platform is capable of restoring real driving scenarios to a certain extent. If the environment changes, the applicability of the method can still be guaranteed.

In this section, experimental steps are introduced. First, the training map to train the personalized driving decision model is selected. Online interactive training is used to converge the model, and output and save training data. When the model reaches the expected training results, training stops and the current model network parameters are saved. Then, 3 test maps (Figure 4) are selected to test the personalized driving decision model. Finally, the test data are analyzed to obtain the learning effect of personalized driving decisions.

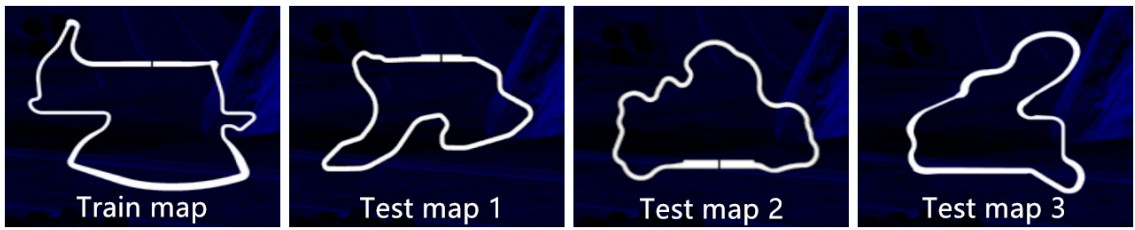

**Figure 4.** Training and testing map.

The track of the training map was approximately 2578 m long, and the lanes were 10 m wide. Straight lines, curves and sections of varying degrees of the curve were included in the track, which is representative. The single-round training steps and driving distance were set to an upper limit to ensure the effectiveness of the training. Immediately after the vehicle runs off the road, the current round of training was terminated. To compare the convergence of the proposed methods, we used the DQN, AC, and A3C algorithms for comparison [29,30]. These algorithms are all classic DRL methods. Since DQN and AC algorithms cannot handle continuous action tasks, we discretize the action space and state space. The loss and reward values can reflect the convergence process of training (Figure 5). For the three algorithms that complete convergence, the moving average of rewards per 10,000 steps can reflect the convergence performance (Figure 6).

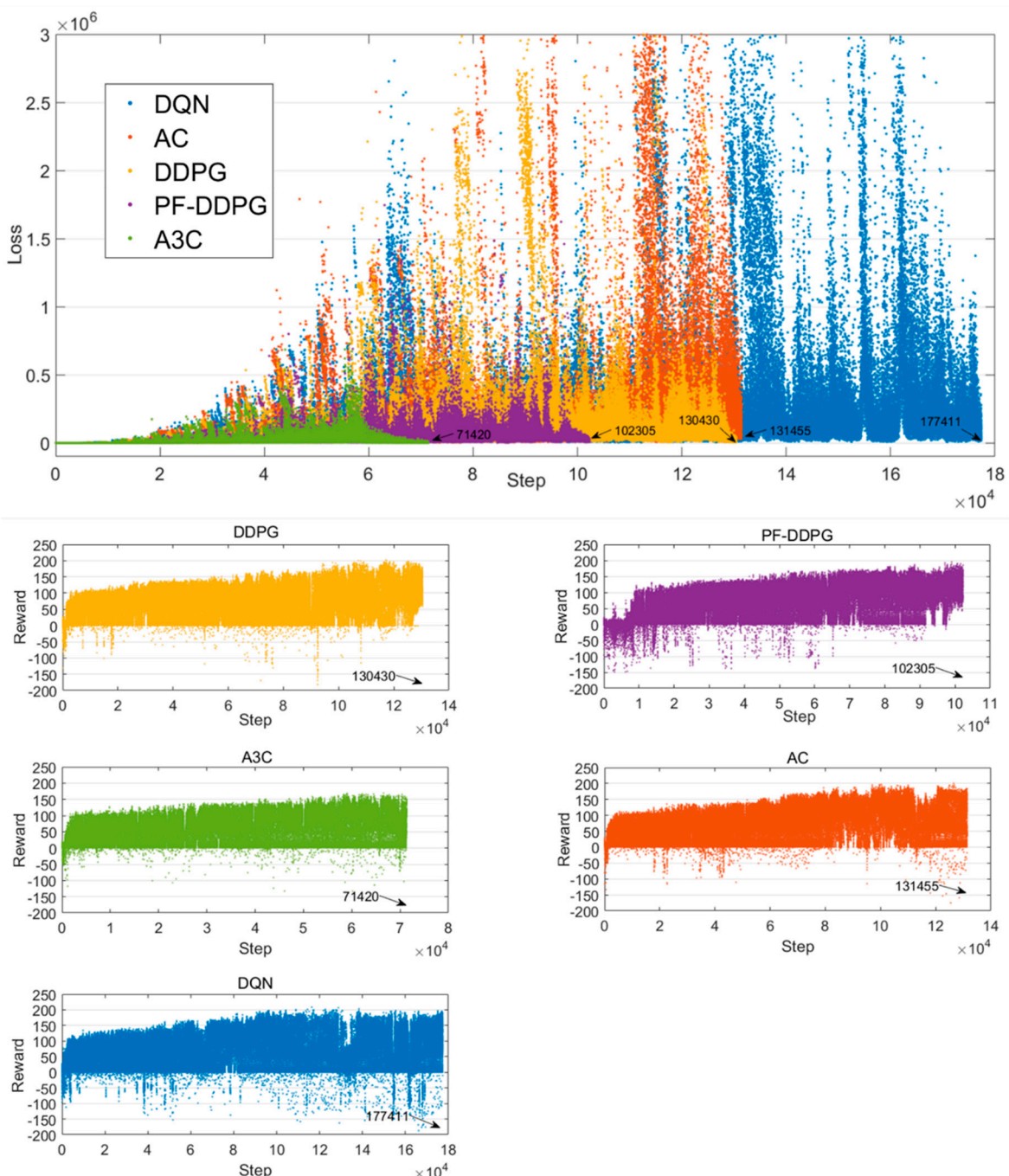

**Figure 5.** Loss value and reward value of the training process.

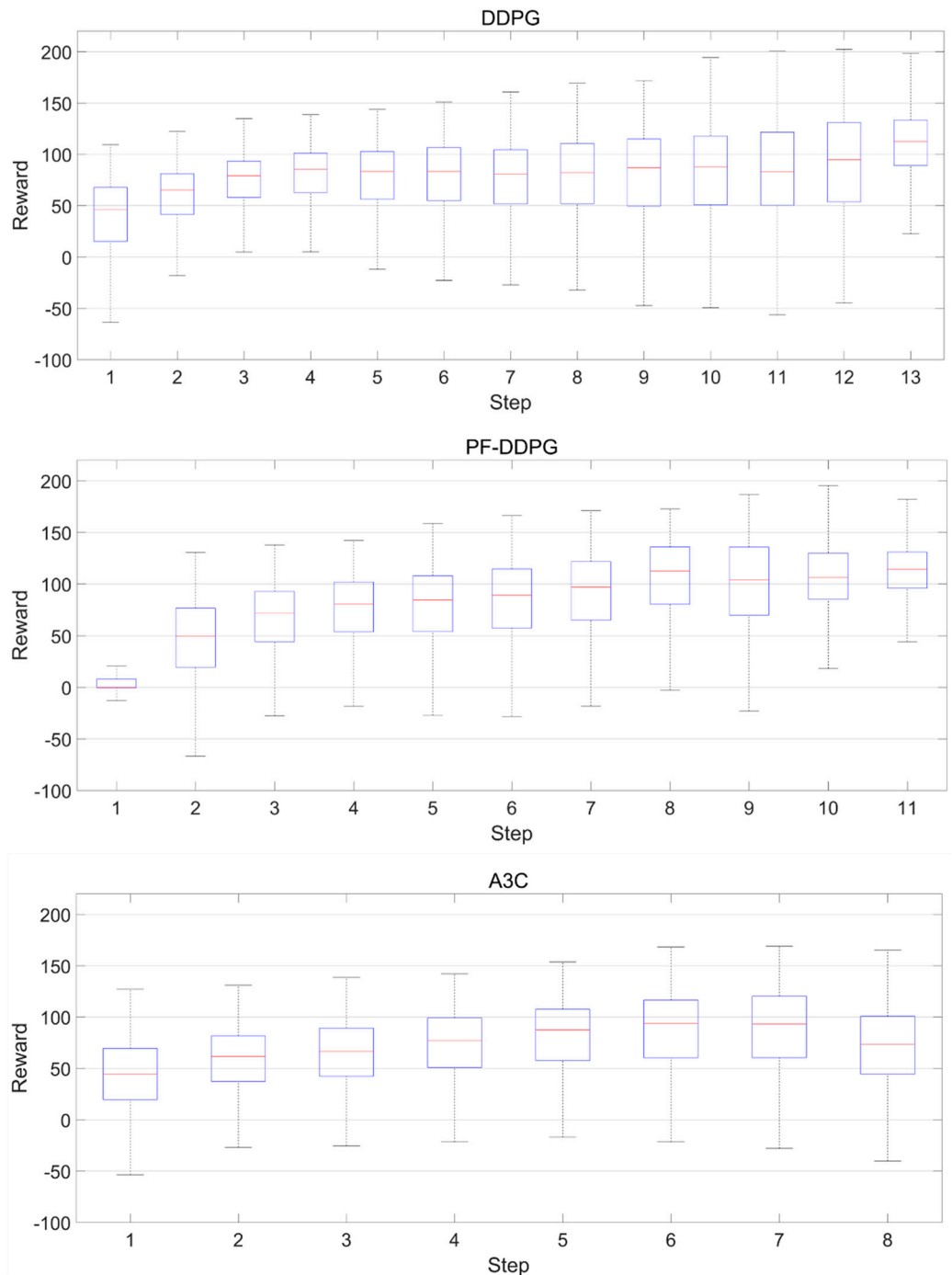

**Figure 6.** Moving averages of the reward values.

For the first 10,000 steps, the model was barely able to obtain positive rewards. During this time, the vehicle collided on the road. The number of steps per episode was very small. The current episode was often terminated because the vehicle ran off the track. As the training progressed and the network was updated, the model iterated toward obtaining higher rewards under the evaluation of the reward function. Starting from approximately 60,000 steps, the model received fewer negative rewards, and the maximum reward for each step also steadily increased. After training 98,000 steps, PF-DDPG converged and always received a positive reward.

To verify the safety and effectiveness of the models, three other maps were selected for testing. Each test map contained straight roads and curves with different curvatures. The proposed method

was tested 5 times on each map, and the data for each test was recorded, including completion time, maximum speed, minimum speed, and the number of damages. Since the total length of each map was different, the completion time was scaled according to the length of the track to the time required to complete 2600 m (training map length) for subsequent data analysis (Table 3). The completion time required an average of 5 tests. To analyze the results of the proposed method in different maps, the training scores and test scores were displayed together.

**Table 3.** Training and testing scores.

| The Proposed Method | Time of Completion/s | Time Difference from Training Map/s | Top Speed/(m/s) | Min Speed/(m/s) | Damage |
|---|---|---|---|---|---|
| **Train Map** | | | | | |
| K = 0.47 | 38.4 | / | 30.1 | 18.7 | 0 |
| K = 1.13 | 34.4 | / | 35.1 | 21.4 | 0 |
| K = 1.91 | 32.2 | / | 36.9 | 24.3 | 0 |
| **Test Map 1** | | | | | |
| K = 0.47 | 41.9 | +3.5 | 30.3 | 17.6 | 0 |
| K = 1.13 | 38.4 | +4.0 | 34.7 | 18.7 | 0 |
| K = 1.91 | 37.1 | +4.9 | 37.2 | 16.8 | 0 |
| **Test Map 2** | | | | | |
| K = 0.47 | 41.2 | +2.8 | 30.4 | 19.6 | 0 |
| K = 1.13 | 37.9 | +3.5 | 35.4 | 23.5 | 0 |
| K = 1.91 | 36.3 | +4.1 | 36.8 | 25.2 | 0 |
| **Test Map 3** | | | | | |
| K = 0.47 | 40.1 | +1.7 | 30.1 | 20.8 | 0 |
| K = 1.13 | 36.8 | +2.4 | 35.6 | 24.5 | 0 |
| K = 1.91 | 34.1 | +1.9 | 37.3 | 25.3 | 0 |

From the test results, PF-DDPG can safely complete the test without collision. Although the time to complete the test maps was longer than the training map, the gap of 2600 m was within 5 s. When the model performed driving tasks on a test map, every decision is a test. Therefore, 3 test maps are 3 test sets. PF-DDPG does not collide in all tests and shows good safety.

To observe the effect of PF-DDPG, first, a piece of data was selected to analyze the throttle opening (Figure 7). Within 50 s, 2 models and drivers completed more than two speed increases. Driver 3 was compared as an example. The throttle opening of Driver 3 was always maintained at a high level. DDPG had a lower throttle opening in the test map than Driver 3, and the difference was obvious. The throttle opening of PF-DDPG improved, which was closer to Driver 3, after adding the personalized factor and training in the same way.

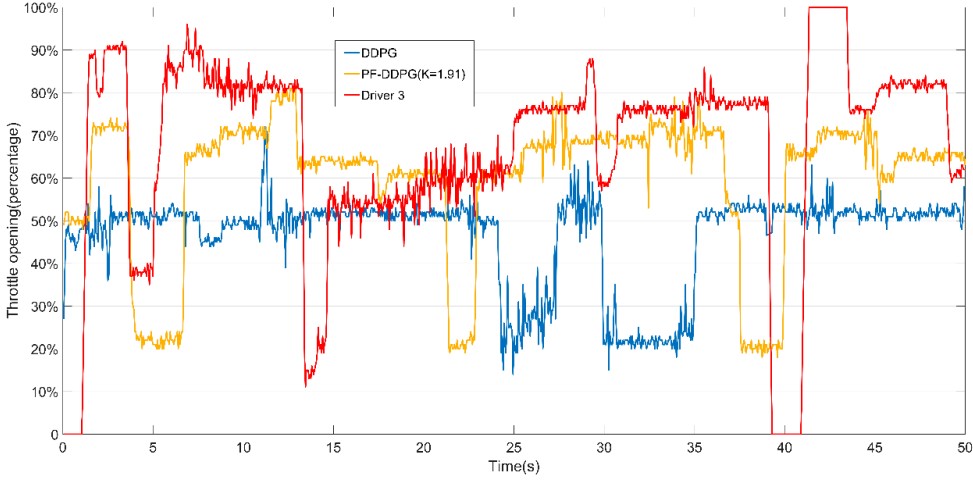

**Figure 7.** Throttle comparison of PF-DDPG, DDPG and Driver 3.

For automatic driving data, speed, steering and throttle opening were selected for comparison with manual driving data. The full data of the 3 drivers were still used as an example.

In terms of driving speed, PF-DDPG showed similar differences to drivers (Figure 8). The distribution of speed was similar to that of the drivers being studied. The degree of data dispersion was surprising. The speed dispersion of the three drivers was very close, and the dispersion of the models was quite different. We determined that this is related to the difference in decision-making mechanisms. In the same driving scenario, the driver's driving personality is limited, but the decision model seeks to obtain the maximum cumulative return while driving. Different parameters can lead to increased driving performance. This also confirms the effectiveness of our algorithm.

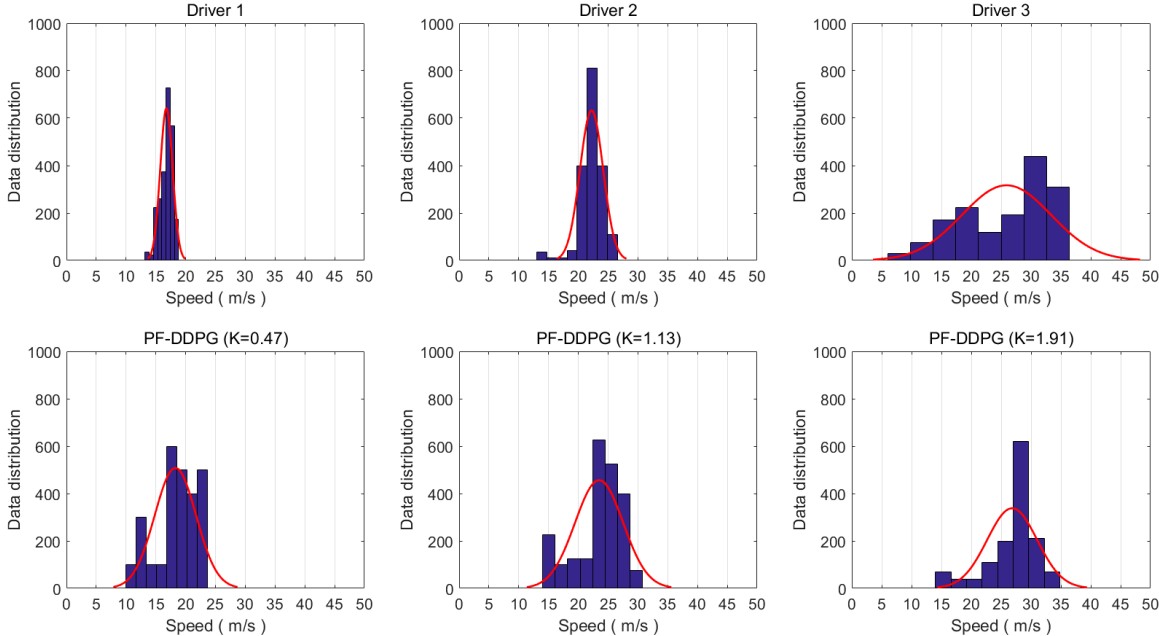

**Figure 8.** PF-DDPG driving speed distribution compared with drivers.

The steering performance of PF-DDPG was also very close to that of drivers (Figure 9). The distribution was most concentrated when the *K* value was 0.47, and the steering operation was the largest when the *K* value was 1.91. Compared to the drivers, the steering operation of the model was more unpredictable. This was caused by environmental uncertainty.

In terms of throttle opening, the individualized decision model was quite different from the drivers (Figure 10). The performance of PF-DDPG was higher than that of the drivers when the *K* value was 0.47 and 1.13. When the *K* value was 1.91, the distribution of PF-DDPG was not dispersed by the data of Driver 3. We analyzed this in relation to the driving strategy adopted by the driver and the algorithm. Drivers sometimes decrease acceleration and let the car coast. However, the model keeps the throttle open. In the case of discreteness, there are also cases where the driving speed was similar. The difference was not as obvious as the drivers. The degree of dispersion of the model was very close.

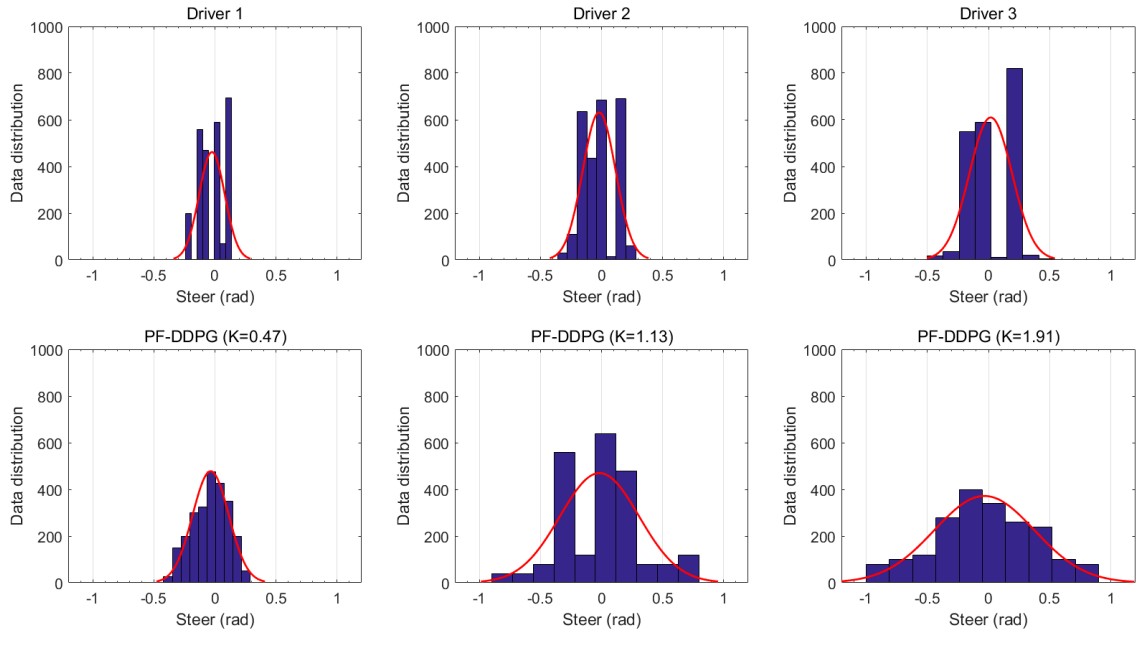

**Figure 9.** PF-DDPG steering action distribution compared with drivers.

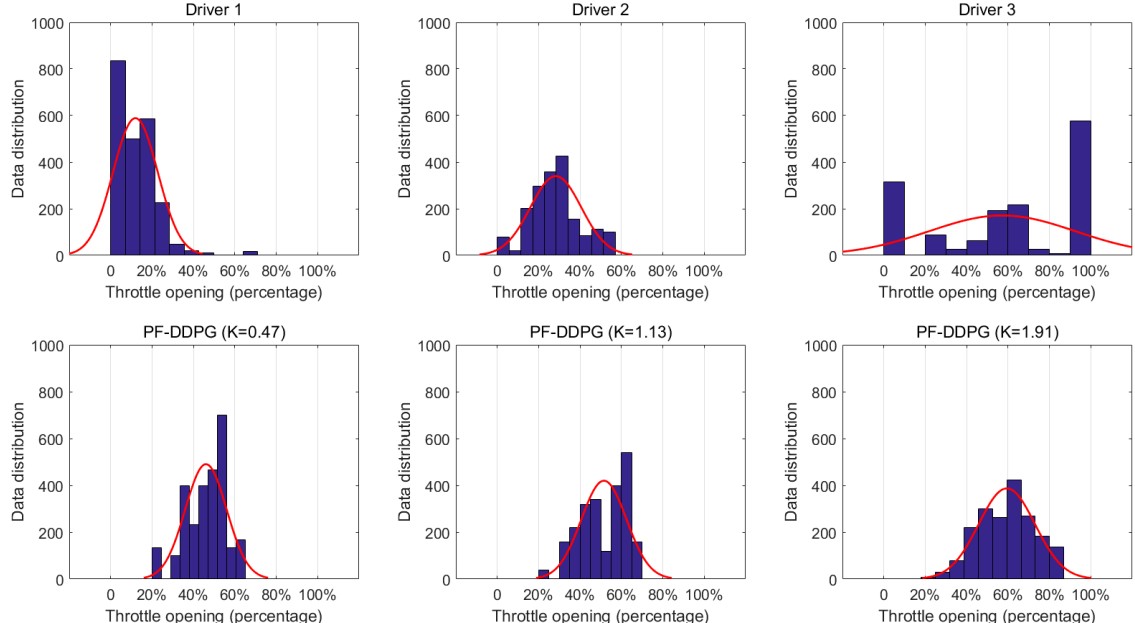

**Figure 10.** PF-DDPG acceleration distribution compared with drivers.

Through data analysis, the behavior of the personalized driving decision model was similar to the drivers in key parameters. Personalized driving decisions were effectively learned by modifying personalized factors. The proposed method shows certain advantages in personalized driving decision learning. Compared with model-based or other methods of autopilot research, the preliminary work of this method is more concise. The amount of time and effort spent on model building and tuning is not needed. The training period is also short, with a greater advantage.

## 4. Discussion

Although the performance of the personalized driving decision model in the test corresponds with expectations, we also identified some problems.

- The proposed method does show differences in personalized indicators. Compared to manual driving data, the difference was not obvious enough. Personalization needs to be improved.
- The personalized driving decision model is somewhat unstable during the acceleration phase. This phenomenon disappears after the speed of the vehicle is stable.
- During training and testing, no other vehicles were added to the road. Since our work is exploratory, the introduction of other vehicles will greatly increase the complexity of the research. In the follow-up work, we will add other vehicles to the scene.

## 5. Conclusions

In this study, a model is designed to validate the feasibility of deep reinforcement learning in the self-learning of personalized driving decisions. Driving data are collected for multiangle, multidimensional analysis. The method of selecting individualized driving indicators is determined, and the key features of the driving characteristics that influence the driving decision are extracted. For the automatic driving task, the driving decision system is designed based on the DDPG algorithm. The environmental parameters are entered, and the driving action parameters are output. Through pretraining, the input driving parameter dimension, the reward function and the number of iterations are determined. The mapping relationship from the sensing parameters to the motion parameters is learned to realize the self-learning of driving decisions. Considering the functional support of the simulation platform, the personalization factor is introduced and associated with the personalized indicator. A personalized driving decision model is built based on the decision-making self-learning ability. The personalized factor is set to the appropriate initial value. Finally, the model is trained and tested online. The advantages of the proposed method are demonstrated by comparing with traditional methods and drivers. It is possible to effectively self-learn personalized driving decisions without too much work.

In future work, we will attempt to solve the problems in the previous section. More powerful neural networks will be considered for use in models such as convolutional neural networks and generative adversarial networks. More attention will be focused on adjusting the hyperparameters. To achieve automatic driving that is closer to reality, research on some of the more advanced driving platforms that include virtual sensors will be considered. Driving environment perception and reconstruction studies will be considered to achieve an experimental environment closer to the actual traffic environment and improve the applicability of the method. To make the model more personalized, the factors of different parameters will be adjusted separately. As a necessity, collision avoidance rules and traffic rules will also be perfected.

**Author Contributions:** Conceptualization, C.W. and Z.C.; Methodology, X.W. and J.X.; Software, X.W.; Resources, C.W.; Data curation, J.X.; Writing—original draft preparation, Z.C. and X.W.; writing—review and editing, C.W. and J.X.; Visualization, X.W.; Supervision, C.W. All authors have read and agreed to the published version of the manuscript.

**Funding:** This work is partially supported by the National Nature Science Foundation of China under Grants U1764262, 61703319, and 51775396, and by National Key R&D Program of China under Grant 2017YFB0102500. the Major Project of Technological Innovation of Hubei Province (2017CFA008).

**Conflicts of Interest:** The authors declare no conflict of interest.

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
