# Peer review of "A Method of Personalized Driving Decision for Smart Car Based on Deep Reinforcement Learning"

_information, doi:10.3390/info11060295_

Round 1

Reviewer 1 Report

Comment to authors:

The authors proposed an interesting method to handle the automatic driving decision task. This topic is on the trend. The content of this manuscript is very easy to follow and understand. The experimental designs also very solid and robust. The outcome is very strong to convince the readers. One of the weaknesses is the literature reviews and methodologies parts.

I think authors have to provide more description on the literature reviews for readers to understand the current progress on this topic.

The authors have to describe the methodologies more tetailedly to improve the clearness of this method.  

Author Response

We really appreciate your suggestions. The new literature review has been added to lines 67 to 88. We also strengthened the description of the method.

Previously, people ignored the potential of deep reinforcement learning. Most of the research on automatic driving based on this method only focuses on safety. The related improvements are not novel. This is also one of the reasons for the lack of literature.

Reviewer 2 Report

This research contains an interesting application using reinforcement learning structures applied in autonomous navigation. However, I found the manuscript can be improved in order to be published.

  1. Some English sentences are not well described. I suggest the authors improve the context of the introduction and conclusion sections.
  2. In the conclusions, there is a sentence "This section is not mandatory, but can be added to the manuscript if the discussion is unusually long or complex". I suggest the authors to eliminate such sentence.

Author Response

We apologize for this sentence in the manuscript. This may be caused by inconvenience in communication, and we have deleted the revised sentences in the new manuscript.

We added a literature review and instructions to the introduction. Discussion and conclusion are separated, and the conclusion part is optimized.

Reviewer 3 Report

The authors propose a method of personalized deriving decision for smart car using deep reinforcement learning. The topic is interesting, and the article fits the scope of the journal. Moreover, the-state-of-the-art provided is well presented. The materials, methodology, results, and discussion are also adequately described. English use and formatting are accurate. I encourage the authors to take into consideration the minor comments provided below.

  • Line 320 and 321: The sentence "However, the difference in speed…." does not read correct.
  • I recommend the authors to keep the conclusions section in the article. Then, do not forget to omit the sentence stated in Line 359 and 360.

Author Response

We really appreciate your suggestions. The 320-line sentence in the original manuscript does not have a clear meaning. We have deleted this sentence.
We apologize for keeping 359 lines of sentences in the manuscript. We have re-checked the manuscript.

Reviewer 4 Report

The discussion section must be separated from the results section and must provide a comparison with other similar methods.

Author Response

We really appreciate your suggestions. The discussion has been separated from the conclusion and is placed in the 360 lines of the new manuscript.
Concerning the comparison with other methods. After we finish our research work, we urgently want to compare with methods with similar functions. Unfortunately, previous research on autonomous driving did not consider personalization, and personalized research lacked the function of automatic driving. We consider this comparison unfair. Therefore, a comparison was made using deep reinforcement learning methods that did not consider personalization (Figure 5).

Round 2

Reviewer 1 Report

Comment to authors:

 I am satisfied with this revision manuscript.

It can be accepted in this time.